# HIV-1 Intasomes Assembled with Excess Integrase C-Terminal Domain Protein Facilitate Structural Studies by Cryo-EM and Reveal the Role of the Integrase C-Terminal Tail in HIV-1 Integration

**DOI:** 10.3390/v16071166

**Published:** 2024-07-20

**Authors:** Min Li, Zhen Li, Xuemin Chen, Yanxiang Cui, Alan N. Engelman, Robert Craigie

**Affiliations:** 1Laboratory of Molecular Biology, National Institute of Diabetes and Digestive and Kidney Diseases, National Institutes of Health, Bethesda, MD 20892, USA; 2Department of Cancer Immunology and Virology, Dana-Farber Cancer Institute, Boston, MA 02215, USA; zhen_li@dfci.harvard.edu (Z.L.); alan_engelman@dfci.harvard.edu (A.N.E.); 3School of Life Sciences, Anhui University, Hefei 230601, China; 21037@ahu.edu.cn; 4Laboratory of Cell and Molecular Biology, National Institute of Diabetes and Digestive and Kidney Diseases, National Institutes of Health, Bethesda, MD 20892, USA; yanxiang.cui@nih.gov; 5Department of Medicine, Harvard Medical School, Boston, MA 02115, USA

**Keywords:** retrovirus, integrase, integration, intasome

## Abstract

Retroviral integration is mediated by intasome nucleoprotein complexes wherein a pair of viral DNA ends are bridged together by a multimer of integrase (IN). Atomic-resolution structures of HIV-1 intasomes provide detailed insights into the mechanism of integration and inhibition by clinical IN inhibitors. However, previously described HIV-1 intasomes are highly heterogeneous and have the tendency to form stacks, which is a limiting factor in determining high-resolution cryo-EM maps. We have assembled HIV-1 intasomes in the presence of excess IN C-terminal domain protein, which was readily incorporated into the intasomes. The purified intasomes were largely homogeneous and exhibited minimal stacking tendencies. The cryo-EM map resolution was further improved to 2.01 Å, which will greatly facilitate structural studies of IN inhibitor action and drug resistance mechanisms. The C-terminal 18 residues of HIV-1 IN, which are critical for virus replication and integration in vitro, have not been well resolved in previous intasome structures, and its function remains unclear. We show that the C-terminal tail participates in intasome assembly, resides within the intasome core, and forms a small alpha helix (residues 271–276). Mutations that disrupt alpha helix integrity impede IN activity in vitro and disrupt HIV-1 infection at the step of viral DNA integration.

## 1. Introduction

A defining feature of HIV-1 replication is the integration of its reverse transcribed RNA genome into human DNA to establish a permanent infection in the target cell. Two distinct chemical steps accomplish DNA integration. In the first step, 3′ end processing, two-nucleotides are removed from each 3′ end of the viral DNA (vDNA); in the second step, DNA strand transfer, a pair of phosphoryl transfer reactions covalently join these 3′ ends to the target DNA via a one-step transesterification reaction [1]. In the case of HIV-1 integration, the sites of strand transfer are separated by five nucleotides on each strand of target DNA, resulting in a five-base-pair duplication flanking the integrated provirus upon repair of the single-strand gaps by cellular enzymes. These reactions occur in the context of nucleoprotein complexes referred to as intasomes [2,3], which consist of a multimer of integrase (IN) bound to two vDNA ends. Intasome is a collective term for the series of stable biochemically assembled nucleoprotein complexes on the integration reaction pathway. A pair of vDNA ends are first synapsed by a multimer of IN to form the stable synaptic complex (SSC) intasome; then, 3′ end processing occurs within the SSC, resulting in the cleaved synaptic complex (CSC). The CSC then captures the target DNA to form the target capture complex (TCC). Finally, a pair of DNA strand transfer reactions complete the insertion of the vDNA into the target DNA; IN remains stably associated with the vDNA in the resulting strand transfer complex (STC).

Retroviruses utilize variable numbers of IN protomers to construct functional intasomes. Whereas foamy viruses use the minimal number of 4 subunits to construct a tetrameric IN intasome [4], β-retroviruses and Maedi-visna virus, a sheep lentivirus, use 8 and 16 subunits, respectively [5,6]. Regardless, intasome function is provided by a conserved intasome core (CIC) structure composed of the catalytic IN subunits held together by a pair of synaptic IN C-terminal domains (CTDs). While the catalytic subunits of foamy virus IN supply the synaptic CTDs, these CTDs are donated by additional IN multimers in other intasome structures, accounting for the higher IN stoichiometries in these complexes [7]. Interestingly, cryo-EM studies of HIV-1 intasomes have revealed multiple oligomeric intasomes [8,9,10,11], demonstrating a high degree of IN plasticity in CIC assembly. This inherent IN heterogeneity has limited the workflow in HIV-1 intasome construction strategies. 

HIV-1 IN consists of three domains spanning 288 amino acids. The N-terminal domain (NTD, residues 1–47) contains a HH-CC motif, which binds one equivalent of Zn^2+^ [12]. The catalytic core domain (CCD, 59–202) contains the catalytic triad DDE motif (D64, D116, E152) and has a fold shared by many polynucleotidyl transferases [13]. The CTD (223–288) consists of five antiparallel beta strands (223–270) adopting an SH3-like fold [14,15]. The CTD is the least conserved of the three domains. The presence of all three domains is required for proper integration functionality. The structures of individual domains of HIV-1 IN were determined by means of X-ray crystallography or NMR spectroscopy by the mid-1990s. However, the first HIV-1 intasome structure was not available until 2017 [8]. Cryo-EM structures of HIV-1 intasomes together with other earlier resolved retroviral intasome structures [5,6,16] illustrated how the IN domains wrap around the two vDNAs, which, together with the synaptic CTDs, form the CIC.

Primate lentiviral intasomes that include HIV-1 and simian immunodeficiency virus from red capped mangabey (SIV_rcm_) have a tendency to form “proto-intasome stacks” of varying length, comprising an octameric repeating unit [9,10,11,17]. In some instances, stacking can help with the cryo-EM studies [17]. However, HIV-1 intasome stacks exhibit high particle orientation bias and an inconsistent thickness of ice. Although HIV-1 intasomes can be purified to sufficient homogeneity for high-resolution cryoEM studies, individual unstacked intasomes are a minor fraction of the population and the assembly and purification process is very time-consuming and inefficient. Here, we reasoned that intasome assembly in the presence of an excess of isolated CTD protein would break the stacks, which are formed by domain swapping between intasomes. This approach dramatically increased the yield of individual intasomes and improved the orientation bias and uniformity of ice thickness, resulting in resolution improvement to 2 Å. The structure elucidated previously unseen residues within the “disordered” C-terminal tail (residues 271–281), which is essential for integration activity both in vitro and in vivo [18]. The improved resolution revealed that the tail makes numerous contacts with other IN domains within the intasome. Single-amino-acid changes in the tail that inhibited integration in vitro were also tested for the effect on integration in the context of HIV-1 in cell cultures. The results support our interpretation that residues within the C-terminal tail play a key role in intasome stability. This strategy will greatly facilitate ongoing efforts to study drug interactions with intasomes and how mutations in IN can confer resistance.

## 2. Materials and Methods

### 2.1. DNA Substrates, Peptides, and Other Reagents

The pre-cut viral long terminal repeat (LTR) DNA substrate U5-25 bp was prepared by annealing U5-25 (5′-AGCGTGGGCGGGAAAATCTCTAGCA) with U5-25R (5′-ACTGCTAGAGATTTTCCCGCCCACGCT), and the substrate U5-69T was made by annealing LTR-29 (5′-AAAAAAAAGTGTGGAAAATCTCTAGCA) with U5-69R (5′-ACTGCTAGAGATTTTCCACACTTTTTTTTTTTTTTTTTTTTTTTTTTTTTTTTTTTTTTT-TTTTTTTTTTTTTTTTTTTTTTTTTTTTTT).

Fluorescent DNA substrates were prepared by attaching 6-FAM fluorophore at the 5′ end of oligonucleotides U5-25 or LTR-29. Trifluoroacetic acid-free peptides were synthesized by GenScript (Piscataway, NJ, USA). 3-(benzyldimethyl-ammonio) propanesulfonate (NDSB, cat. # 17236), polyethyleneimine (PEI, cat. # 408719), dimethyl sulfoxide (DMSO, cat. # D8418), efavirenz (EFV, cat # SML0536), and polyethylene glycol 6000 (PEG, cat. # 81253) were purchased from Sigma.

### 2.2. Protein Preparation

His-tagged Sso7d-IN and its C-terminal truncations were expressed and purified essentially as described in [19]. Briefly, IN was expressed in *E. coli* BL21(DE3) and the cells were lysed in lysis buffer (LB, 20 mM Hepes pH 7.5, 10% glycerol, 2.0 mM 2-mercaptoethanol and 1.0 M NaCl) supplemented to contain 20 mM imidazole. The protein was purified by means of nickel-affinity chromatography and the His-tag was removed with thrombin. Aggregated protein was removed by means of gel filtration on a HiLoad 26/60 Superdex-200 column (Cytiva cat # 28989335) equilibrated with 20 mM Hepes pH 7.5, 10% glycerol, 0.5 M NaCl, 2.0 mM 2-mercaptoethanol and 1.0 mM EDTA. His-tagged LEDGF-IBD (residues 307–460) and IN-CTD (residues 185–288) were expressed and purified in essentially the same way, except the His-tags were not removed. Briefly, cells were lysed in LB with 20 mM imidazole and the lysates were loaded onto HisTrap HP columns (Cytiva cat # 17524801), then the columns were washed with lysis buffer supplemented to contain 40 mM imidazole. The His-tagged proteins were eluted from the column with a gradient of imidazole from 0 to 750 mM in LB. Aggregated protein was removed by means of gel filtration on a HiLoad 26/60 Superdex-200 column equilibrated with 20 mM Hepes pH 7.5, 10% glycerol, 0.5 M NaCl, 2.0 mM 2-mercaptoethanol and 1.0 mM EDTA. Purified proteins were concentrated using a centrifugal concentrator (Ultracel-3K, Millipore cat # UFV900308) and flash frozen with liquid nitrogen. Wild-type IN (without Sso7d fusion) and its C-terminal truncations were prepared as described [11].

### 2.3. In Vitro Concerted Integration Assay

The concerted integration assay with Sso7d-IN was performed essentially as described in [11,19]. Briefly, Sso7d-IN and U5-25bp vDNA substrate were preincubated on ice in 20 mM Hepes (pH 7.5), 25% glycerol, 10 mM DTT, 5.0 mM MgCl_2_, 4.0 μM ZnCl_2_, 50 mM 3-(Benzyldimethyl-ammonio) propanesulfonate, and 100 mM NaCl in a 20 µL reaction volume. Three hundred nanograms of target plasmid DNA pGEM-9zf was then added and the reaction proceeded at 37 °C for 2 h. The integration product DNA was recovered by means of ethanol precipitation and subjected to electrophoresis in a 1.5% agarose gel in 1× tris-boric acid–EDTA buffer. DNA was visualized either via ethidium bromide staining or via fluorescence using a Typhoon 8600 fluorescence scanner (Cytiva, Marlborough, MA, USA). Concerted integration assay conditions with wild-type IN (WT-IN) were similar to those described previously [11]. Briefly, 0.4 μM of WT-IN and 0.2 μM of U5-69T DNA substrate were incubated in 20 mM HEPES pH 7.5, 12% dimethyl sulfoxide, 10% polyethylene glycol 6000, 5 mM DTT, 10 mM MgCl_2_, 4 μM ZnCl_2_, 100 mM NaCl, and 300 ng of pGEM-9zf in a 20 μL reaction volume, and the reaction was carried out at 37 °C for 2 h.

### 2.4. Intasome Assembly and Electrophoretic Mobility Shift Assays

Intasomes were assembled essentially as described in [11]. Briefly, 3.0 μM Sso7d-IN proteins and 1.0 μM U5-25bp vDNA substrate were preincubated on ice in 20 mM Hepes (pH 7.5), 25% glycerol, 10 mM DTT, 5.0 mM MgCl_2_, 4.0 μM ZnCl_2_, 50 mM 3-(Benzyldimethyl-ammonio) propanesulfonate, 50 μM dolutegravir (DTG), and 100 mM NaCl in a 20 µL reaction volume. The reaction was carried at 37 °C for 2 h. Heparin (10 μg/mL final concentration) was added to the assembly reaction and a 2.5 μL aliquot was subjected to electrophoresis on a 3.0% low melting agarose (SeaKem LE agarose) containing 10 μg/mL heparin in 1× tris-boric acid–EDTA buffer. DNA was visualized by means of fluorescence using a Typhoon 8600 fluorescence scanner.

### 2.5. Hetero-Intasome Assembly and Purification for Cryo-EM

The CSC intasome assembly was performed essentially as described in [9,11]. Briefly, 3.0 μM Sso7d IN, 15.0 μM CTD, and 1.0 μM U5-25bp viral DNA substrate were incubated in 20 mM Hepes (pH 7.5), 25% glycerol, 5 mM 2-mercaptoethanol, 5.0 mM MgCl_2_, 4.0 μM ZnCl_2_, 50 mM 3-(Benzyldimethyl-ammonio) propanesulfonate, 50 μM DTG, and 100 mM NaCl at 37 °C for 2 h. We note that all the intasomes used for integration activity assays were assembled in the presence of 5.0 mM CaCl_2_ and DTG was omitted from the reaction buffer. The reaction was stopped by transferring the mixture to ice, and 500 mM NaCl (final) was added to the assembly reaction mixture. To increase the solubility of the intasomes, 6.0 μM LEDGF-IBD (residues 307–460) was added to the reaction and incubated at room temperature for 2 h. Intasomes were first purified by a HisTrap column equilibrated with 20 mM Tris-Cl pH 8.0, 5 mM 2-mercaptoethanol, 0.5 M NaCl, and 20% glycerol. Eluted intasomes and free proteins were then separated by means of gel filtration on a Superose 6 Increase 10/300 GL column (Cytiva cat # 29091596) equilibrated with 20 mM Tris-HCl pH 6.2, 0.5 mM TCEP, 500 mM NaCl, 5 mM MgCl_2_, and 6% glycerol. Intasomes corresponding to the monodispersed species were collected for cryo-EM.

### 2.6. Cryo-EM Grid Preparation and Data Acquisition

Purified CSCs (3.0 μL of 0.4 mg/mL) were applied onto freshly glow-discharged (30 s, PELCO easiGlow) holey gold grids (UltrAuFoil R 1.2/1.3 300 mesh, QUANTIFOIL), adsorbed for 10 s, blotted for 4 s with force 4, and then plunged into liquid ethane using a Vitrobot plunge freezer (Thermo Fisher Scientific) at 20 °C with 100% humidity. Micrographs were acquired using SerialEM [20] installed on an FEI Titan Krios G3 electron microscope operating at 300 kV. Movies were recorded on a Gatan K3 Summit direct electron detector at 105 K nominal magnification in super-resolution mode (corresponding to a calibrated pixel size of 0.415 Å at the sample level), with defocus values ranging from −0.6 to −2.6 μm.

### 2.7. Structure Determination and Refinement

Beam-induced movement in each movie was corrected and aligned using MotionCor2 [21] with dose-weighting applied and the pixels were binned 2 to 0.83 Å. Ref. [22] methodology was used to estimate the contrast transfer function. Intasome particles were picked using Gautomatch (developed by Dr. K. Zhang; https://sbgrid.org/software/titles/gautomatch, accessed on 26 June 2024) from the dose-weighted micrographs and used to create an initial raw particle stack. 2D, 3D classification and auto-refinements were carried out in RELION v3.0 [23]. Selected particles were further refined with contrast transfer function (CTF) refinements and Bayesian polishing. Final auto-refinement resulted in a map with 2.0 Å resolution with a CIC mask applied. DeepEMhancer [24] was used to reduce noise in the final map. The detailed data processing and flow-chart are shown in Appendix A. Resolution was based on the “gold standard” refinement procedure and the 0.143 Fourier shell correlation (FSC) criterion [25]. Local resolution was estimated using ResMap [25]. UCSF Chimera [26], Coot [27], PHENIX Real-Space refinement [28], and MolProbity [29] were used for model building, refinement, and validation. Previous cryo-EM structures (PDB 5U1C, 8W34) facilitated the model building. UCSF Chimera and PyMOL (Pymol.org) were used to make structural figures.

### 2.8. Virus-Based Work

#### 2.8.1. HIV-1 Production and Infection

HEK293T cells obtained from the America Type Culture Collection (cat. # CRL-3216) were grown in Dulbecco’s modified Eagle’s medium containing 10% fetal bovine serum, 100 IU/mL penicillin, and 100 μg/mL streptomycin (DMEM) at 37 °C in the presence of 5% CO_2_. Viruses were generated by co-transfecting 10 µg of total DNA consisting of pNLX.Luc.R- and VSV-G expression construct [30] at a 6:1 ratio onto 4 × 10^6^ cells plated in 10 cm dishes the previous day using PolyJet™ DNA transfection reagent (SignaGen Laboratories cat. # SL100688) according to the manufacturer’s protocol. Two days post transfection, virus-containing cell supernatants were collected and centrifuged at 500× *g* for 5 min to remove cell debris before filtration via gravity flow through 0.45 µm filters and treatment with 20 U/mL Turbo DNase (Invitrogen cat. # AM2238) for 1 h at 37 °C to remove residual plasmid DNA. Virus concentration was assessed using technical duplicate samples with p24 ELISA (Advanced Bioscience Laboratories cat. # 5447).

Infections normalized for p24 content (0.25 pg/cell) were performed using 2 × 10^5^ HEK293T cells per well of a 24-well plate (1 mL). After 6 h at 37 °C, the medium was replaced with fresh DMEM. At 48 h post infection, cells were lysed using passive lysis buffer as recommended by the manufacturer (Promega cat. # E194A). Cell lysates were clarified by means of centrifugation at 21,100× *g* for 15 min at 4 °C, and luciferase activity was determined and normalized to total protein levels in cell extracts using technical duplicate samples as previously described [31].

#### 2.8.2. Quantitative PCR Assays

Infections were performed as described above and quantitative PCR (qPCR) assays were conducted essentially as previously described in [32]. In brief, cells were harvested at 8 h post infection to assess late reverse transcription (LRT) products, while the levels of Alu integration and 2-LTR circles were assessed at 24 h post infection. Parallel infections were conducted in the presence of 20 µM EFV to account for any residual plasmid DNA that may have persisted in virus stocks, and qPCR values obtained in EFV-treated samples were subtracted from experimental LRT and 2-LTR circle samples. DNA was extracted from cells using the Quick-DNA™ Microprep Kit (Zymo cat. # D3021). DNA quantities were normalized by means of spectrophotometry and 10 ng (Alu integration), 25 ng (LRT), or 200 ng DNA (2-LTR circles) of technical triplicate samples were analyzed via qPCR. LRT and 2-LTR Cq values were compared to dilutions of plasmid molecular clones as described in [32]. Following the first round of PCR, 2 µL of a 1:100 dilution of first round reaction products was analyzed by means of a second real-time step for Alu integration. Integration levels were percent-normalized to WT samples after subtracting control values that were determined by omitting the Alu-specific primer from parallel first-round PCR assays.

## 3. Results

### 3.1. CTD Stimulates the Full Length IN Integration Activity

Intasome stacks are the dominant population of intasomes assembled in vitro for both HIV-1 [9,11,17] and SIV_rcm_ [17]. The CTDs of the CIC are donated from different protomers by domain swapping in the stacks, resulting in an octameric intasome repeating unit, each with a pair of vDNA ends. In the individual intasomes, the CTDs are donated from flanking integrase protomers (Figure 1). Stacks are not biologically relevant, as only a single pair of vDNA ends are available in infected cells. However, the abundance of stacks in vitro presents difficulties with cryo-EM data collection and analysis. Herein, we asked whether supplying an excess of isolated CTDs in trans might “break” the stacks (Figure 1). We refer to CSC intasomes with an external source of CTDs as hetero-intasomes (or hetero-CSCs).

Approximately 50 amino acid residues comprise the 5-stranded β sheet fold of the CTD [14,15]. However, we initially tested the effects of isolated C-terminal fragments of different lengths on concerted DNA integration. Residues 185–288, which encompass the β sheet fold (residues 223–270), showed the greatest stimulation and were selected for detailed study; we define the 185–288 protein herein as the CTD for simplicity. Increasing concentrations of the CTD stimulated concerted integration with both Sso7d-IN (Figure 2A) and WT-IN (Figure 2B). Maximal stimulation occurred at a CTD:IN ratio of about 5:1 for Sso7d-IN and 10:1 for WT-IN. The mechanism of concerted integration stimulation by the CTD is unclear, but unlike other peptides that stimulate integrase protein activity in vitro [33], this domain participates directly in intasome assembly (as shown below).

### 3.2. Exogenous CTD Can Be Incorporated into Intasomes

Stimulation of concerted integration activity suggested that isolated CTDs can be incorporated into intasomes. However, it remained possible that stimulation of activity resulted from indirect effects without evidence of incorporation, as has been observed with P5 peptide [33].

Heterointasomes were assembled by mixing full-length Sso7d-IN, His-tagged CTD, and vDNA, essentially as depicted in Figure 1C. Previous lentiviral intasome studies have leveraged the IN host cofactor lens epithelium-derived growth factor (LEDGF) [6,8,9,10,11,17]. In addition to full-length LEDGF, the P5 peptide derived from its AT-hook region as well as its IN-binding domain (IBD) have facilitated intasome assembly and solubility. We too used LEDGF-IBD (residues 307–460) to improve intasome solubility. Following assembly, the intasomes were crudely purified by a HisTrap Ni column followed by a gel filtration column to remove unreacted vDNA and proteins. The hetero-CSCs were eluted as a major peak on a Superose 6 column (Figure 3A). SDS-PAGE analysis of the elution fractions confirmed that the CTD as well as the IBD were indeed incorporated into the intasomes, which eluted from 1.4 mL to 1.9 mL (Figure 3B). The purified intasomes were competent for concerted integration activity (Figure 3C). Notably, intasomes assembled in the presence of the IBD and CTD were as active as complexes assembled with just the added IBD or with Sso7d-IN without added protein cofactors (Figure 3D). Semi-quantitative analysis of the hetero-CSC stoichiometry suggested that the protein ratio of full-length IN:IBD:CTD was roughly 1:1:0.3 (Figure 3E).

### 3.3. Reconstruction of the Hetero-CSC Intasome Core by Cryo-EM

Early strategies to isolate intasomes that were sufficiently monodispersed for cryo-EM required multiple size exclusion chromatography steps, and the yield was low [8]. In the current protocol, the mono dispersed hetero-CSCs could be isolated directly from a one-step Superose 6 gel filtration column with high yield (Figure 3A). Overall, the particles on the grids were well dispersed with minimal stacks and aggregates, resulting in uniform thin ice (Figure 4A). The workflow of the cryo-EM analysis of hetero-intasomes is shown in Appendix A. After CTF refinement and Bayesian polishing, the CIC map was refined to 2.01 Å (Figure 4B, Appendix A), a resolution that readily accompanied the building of an atomic model (Figure 4C). The resolution map is shown in Figure 4D. As expected for a 2 Å cryo-EM map, aromatic residues tryptophan 61, tyrosine 83, and phenylalanine 185 in the IN CCD showed readily discernable side chains with ring hole features (Figure 4E). DTG, Mg^2+^, and water molecules coordinated in the DDE active site were also unambiguously modeled (Figure 4F).

### 3.4. The C-Terminal Tail of HIV-1 Integrase Stabilizes the Intasome Structure

It has been shown that the C-terminal tail 18 residues (270–288, Figure 5A) of HIV-1 IN are critical for virus integration both in vitro and in vivo [18,34], but its molecular function has remained unclear and structural information for these 18 residues is not available in the context of HIV-1 intasomes. We were able to build an atomic model of the CTD tail of distal IN protomers up until residue 281 with the high-resolution cryo-EM map (Figure 5B). The C-terminal 7 residues of IN were disordered and not visible. The distal CTD and its tail resided within the intasome core, and the tails were largely buried within the assembly (Figure 5C). Residues 271 to 276 were previously visualized as a disordered coil in the absence of vDNA [35]. These residues formed a small alpha helix in our previous intasome structures [9,10,11]; however, the poor density of the cryo-EM maps in this region resulted in an incomplete model, and the functional role of these residues in integration could not be assessed. At the current resolution, we could readily discern that the C-terminal tail engages in a number of interactions with other IN protomers within the intasome (Figure 5D). Tyr271 and Tyr194 (Chain A) maintained a hydrophobic pocket; Lys273 was oriented for potential interaction with vDNA (Chain F) through long-range electrostatic effects; and Gln274 and Gln209 (Chain B) formed a hydrogen bond. Asp279 and Arg269 (Chain C) formed a salt bridge, stabilizing the tail alpha helix structure (Figure 5E). These observations indicated that CTD tail bridging interactions with other IN/vDNA protomers, including chains A, B, C, and F, likely play an important role in intasome stabilization (Figure 5E).

Functional analysis of C-terminal tail deletion mutant constructs showed that the residues 271–279 are critical for concerted integration and directly participate in intasome assembly (Figure 6). For completeness, we constructed C-terminal tail deletions of both WT-IN (Figure 6A) and Sso7d-IN (Figure 6B). IN_1-279_ exhibited the same level of concerted integration activity as the full-length 1–288 proteins. Mutants with progressive tail deletions yielded a dramatic fall off in activity. The small alpha helix composed of residues 271–276 was indispensable for concerted activity (Figure 6A,B). To assess intasome assembly, we performed EMSA with the Sso7d set of proteins. In general, the mutants that failed to support concerted integration activity also failed to form stable intasomes (Figure 6B,C). These results are well aligned with the previous biochemical and viral infectivity studies [18]. The CTD only moderately stimulated the concerted integration activity of the tailless C-terminal truncation mutants IN_1-269_ and IN_1-270_ (Appendix A), which contrasts with the stimulation for full-length IN activity (Figure 2A), suggesting that the distal C-terminal tail cannot readily be incorporated in trans. Similarly, peptides of the C-terminal tail had minimal impact on full-length Sso7d-IN activity, with comparatively high concentration of the WT tail sequence suppressing concerted integration activity (Appendix A).

To further assess the importance of IN C-terminal tail interactions noted in the structure (Figure 5D,E) on IN activity, we targeted individual amino acid residues by site-directed mutagenesis. Single-amino-acid mutagenesis of the C-terminal tail validated the important interactions found in the structure analysis above. The mutants Y271R and A276P, which predictably disrupted the alpha helix, were defective for concerted integration activity. The Q274L mutant, which would theoretically break the Q274-Q209 tail-Chain B hydrogen bond, the G277Q mutant, for which the bulky glutamine side chain is expected to disrupt the integrity of the alpha helix structure, and the D279R mutant, which would disrupt the D279-R269 tail-Chain C salt bridge, displayed reduced levels of concerted integration activity. By contrast, D279E, which could maintain the salt bridge interaction with R269 (Chain C), displayed the WT level of concerted integration (Figure 6D).

### 3.5. IN Tail Mutants Y271R and A276P Are Defective for Integration during HIV-1 Infection

We next tested the IN C-terminal tail mutants that reduced concerted integration activity in vitro, including Y271R, Q274L, A276P, G277Q, and D279R (Figure 6D), in the context of HIV-1 infection. Changes were introduced into the single-round reporter construct pNLX.Luc.R- that expresses firefly luciferase in place of viral *nef* and is defective for envelop glycoprotein expression [30]. HIV-Luc virus particles were pseudotyped with the heterologous vesicular stomatitis virus (VSV) glycoprotein G via cotransfection, and p24-normalized levels of WT and IN mutant HIV-Luc were used to infect HEK293T cells. Infection, determined via luciferase activity at 2 days post-infection, required the expression of the luciferase reporter gene following viral entry, reverse transcription, and integration. For comparison, we included two different types of defective IN mutant viruses, known as class I and class II. Class I mutants, typified by conservative substitutions of DDE catalytic triad residues (D64N/D116N was used herein), are predominantly defective for integration [36]. By contrast, class II IN mutants display pleiotropic replication defects [36]. Herein, we employed the H12N substitution within the NTD zinc-binding HH-CC motif, which destabilizes virion IN proteins and perturbs proper HIV-1 maturation and downstream steps of reverse transcription and integration [37,38].

Most C-terminal tail IN mutant viruses were released from transfected cells in a similar manner to WT HIV-Luc. The approximate 25% and 40% reductions for Y271R and A276P, respectively, were less pronounced than the approximate 3-fold reductions observed with IN mutant control viruses (Figure 7A). D279R infected cells at a level that was indistinguishable from WT HIV-Luc. While Q274L and G277Q yielded modest ~23% and 30% reductions in HIV-Luc infection, respectively, Y271R and A276P displayed more pronounced ~77% and 71% reductions, respectively. As the class I and class II IN mutant control viruses were much more defective (≥500-fold infection defects), it is obvious that IN C-terminal residues play less critical roles in HIV-1 infection than the residues that comprise the highly conserved HH-CC and DDE motifs (Figure 7B).

To assess the nature of the infection defects, reverse transcription was measured by means of qPCR. We chose to amplify a region of the HIV-1 genome that requires the second template switch of reverse transcription, and is hence referred to as late reverse transcription products or LRT [39]. All IN C-terminal tail mutant viruses synthesized LRT products at levels that were statistically indistinguishable from the levels observed for WT and D64N/D116N. H12N, as expected [38], displayed only ~7% of WT LRT activity (Figure 7C).

Comparing the above infection and LRT results suggested that Y271R and A276P were predominantly defective for integration. To directly assess integration, we performed an Alu integration assay that amplified sequences between HIV-1 R within the LTR and cellular Alu sequences. As expected, the D64N/D116N mutant control was highly defective (>1000-fold) for integration. Y261R and A276P displayed ~80% and 71% reductions in integration, respectively (Figure 7D). Thus, defective integration seemingly fully accounted for the Y271R and A276P infection defects (comparing the results of Figure 7B,D). To verify this conclusion, we assessed levels of 2-LTR circles, which are a byproduct of defective integration. Because class I IN mutant viruses are predominantly defective for integration, they display 2-LTR circle levels in excess of WT HIV-1. Class II IN mutants, due to the defects in reverse transcription, by contrast display reduced levels of 2-LTR circles compared to WT [37]. While the D64N/D116N IN mutant formed an excess of 2-LTR circles of ~13-fold compared to WT, Y271R and A276P displayed ~3.8 and 1.8-fold increases in 2-LTR circles (Figure 7E). The lower levels of 2-LTR circle increase observed for Y271R and A276P compared to D64N/D116N reflects the different extents of integration defect for the viruses (Figure 7D,E). Our data support the conclusion that Y271R and A276P IN mutant viruses are ~5- and 3.5-fold defective for infection due to specific defects in HIV-1 integration.

## 4. Discussion

The HIV-1 IN CTD tail is essential for DNA integration both in vitro and in vivo [18,34]. Although its role remains unclear, the CTD has been implicated in interactions with other cellular and viral proteins (reviewed in [40]) and a disordered tail could be a prime candidate for mediating such interactions. However, this cannot account for the absence of DNA integration activity in vitro when the C-terminal tail is missing. Our results show that the C-terminal tail plays a key role in the structural integrity of HIV-1 intasomes, although other additional roles cannot be excluded. This interpretation is supported by our in vitro and in vivo analyses of the effects of single-amino-acid changes in residues within the tail that form interactions with other domains within the intasome structure.

Which IN protomers can be substituted by the isolated CTD in the dodecameric intasome? Experimentally, the ratio of full-length IN:CTD is approximately 1:0.3, as determined by semi-quantitative Coomassie staining. The flanking dimers in the hetero-intasomes display comparatively weak density compared to intasomes assembled with full-length IN only. In contrast, the density of the remaining part of the hetero-intasome structure is comparatively well resolved (Appendix A). Whereas full-length IN normally donates both proximal and distal CTD dimers, we conclude that the flanking dimer domains can be substituted by the CTDs added in trans. This explains why the isolated CTDs can break stacks, as the synaptic CTD is contributed by the flanking dimers and the synaptic CTD is the domain that is involved in stack formation by domain swapping.

Integrase strand transfer inhibitors (INSTIs) are now front-line drugs for the treatment of HIV [41]. These drugs target intasomes, rather than free IN protein. Although the newer INSTIs have an improved resistance profile, resistance remains a problem. High-resolution structural studies of intasomes are therefore required to understand their mechanisms of action and how mutations in IN can confer resistance. Although considerable progress has been made, such studies have been difficult and time consuming because only a minor fraction of the assembled intasomes exist as single entities, with the majority forming stacks. The finding that the population of single intasomes can be greatly increased by breaking the stacks using isolated CTDs will facilitate ongoing studies of the interaction between drugs that target intasomes and the mechanisms by which IN can develop resistance.

## Figures and Tables

**Figure 1 viruses-16-01166-f001:**
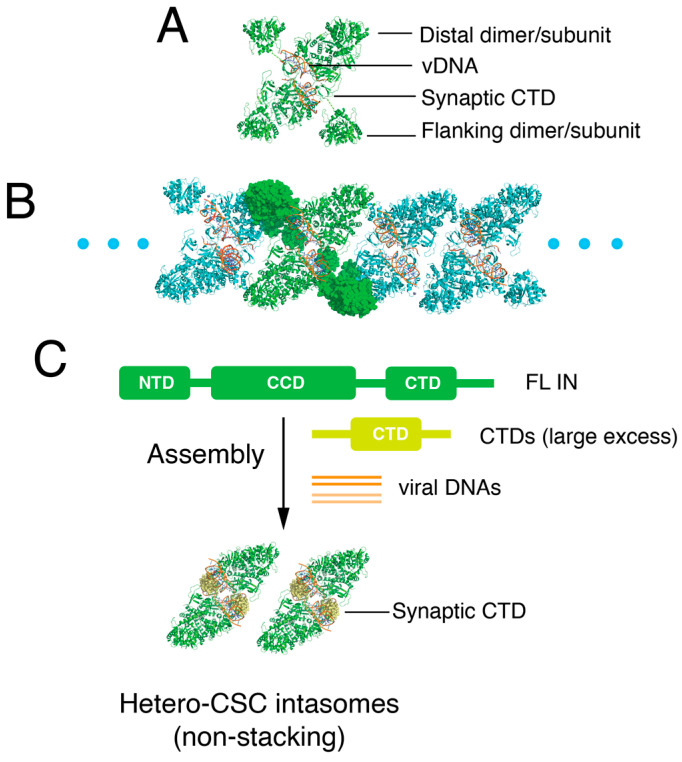
Schematic of “hetero-intasome” assembly. (**A**) Domain organization of dodecameric HIV-1 intasomes. Green, IN proteins; red, vDNA. (**B**) HIV-1 “oligomer-intasome”, which is a stack of octameric intasomes shown as cartoons and space fill (flanking subunit). One dodecameric unit is shown in green, and the other repeat units are shown in cyan. (**C**) Assembly of “hetero-intasomes” with an excess of CTD in the assembly reaction mixture. Exogenous synaptic CTDs are shown as space-fill in light green.

**Figure 2 viruses-16-01166-f002:**
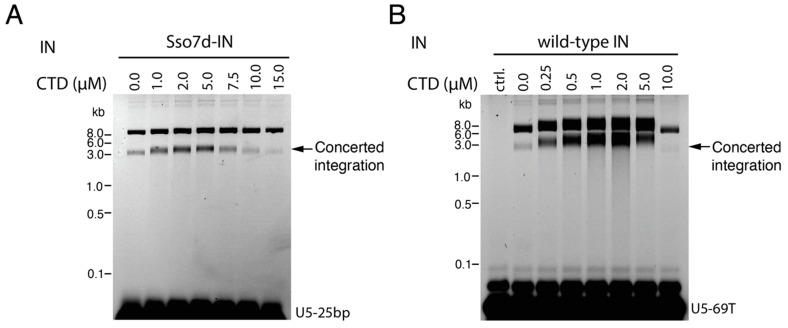
Excess CTD stimulates full-length IN integration activity. (**A**) Different amounts of CTD were preincubated with Sso7d-IN and pre-cleaved U5-25bp vDNA substrate prior to initiation of strand transfer at 37 °C. Recovered DNAs were visualized by fluorescence using a Typhoon 8600 scanner. (**B**) Indicated amounts of CTD protein were preincubated with WT-IN and pre-cleaved U5-69T DNA substrate prior to incubation at 37 °C for 2 h. DNA integration products were analyzed as in panel A. The concentrations of CTD and concerted integration products are indicated. Molecular mass standards in kb are indicated to the left of the gel panels. Ctrl, negative control sample that omitted WT-IN from the strand transfer assay.

**Figure 3 viruses-16-01166-f003:**
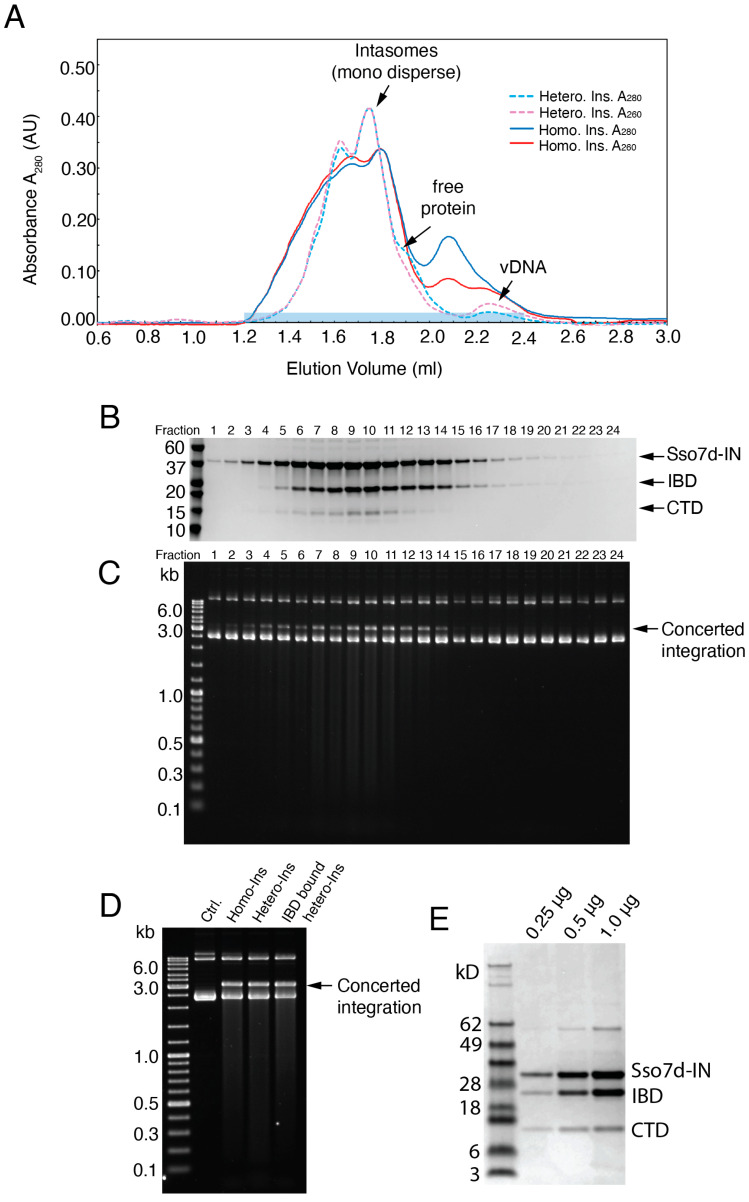
Size-exclusion chromatography and integration activity of hetero-intasomes assembled with full-length Sso7d-IN and CTD. (**A**) Elution profile of hetero-intasomes on the Superose 6 Increase 10/300 GL column (Cytiva). Monodispersed hetero-intasomes and free proteins and vDNA are indicated with arrows. Homo-intasomes formed with Sso7d-IN without added CTD are shown for comparison. (**B**) Fractions 1 to 24 (lane 1 to 24), corresponding to 1.2 mL to 2.4 mL elution volumes, were analyzed by means of SDS-PAGE. Sso7d-IN, LEDGF-IBD, and CTD are indicated with arrows. Mass standard positions (in kD) are indicated on the left. (**C**) Fractions 1 to 24 were assessed for strand transfer activity and integration products were detected by means of ethidium bromide staining. Concerted integration products are indicated on the right and mass standards (in kb) are given on the left. (**D**) Concerted integration activity was not affected by either IBD binding or CTD incorporation. Purified 2.5 nM intasomes assembled with Sso7d-IN (Homo-Ins), Sso7d-IN with CTD (Hetero-Ins), and Sso7d-IN with CTD and IBD (IBD-bound Hetero-Ins) were tested for concerted integration. The negative control (Ctrl.) was target DNA in the absence of added intasomes. (**E**) Purified hetero-intasomes were analyzed by means of SDS-PAGE. Lane 1, 0.25 μg; Lane 2, 0.5 μg; and Lane 3, 1.0 μg, were loaded onto the gel. The results are representative of three independent experiments.

**Figure 4 viruses-16-01166-f004:**
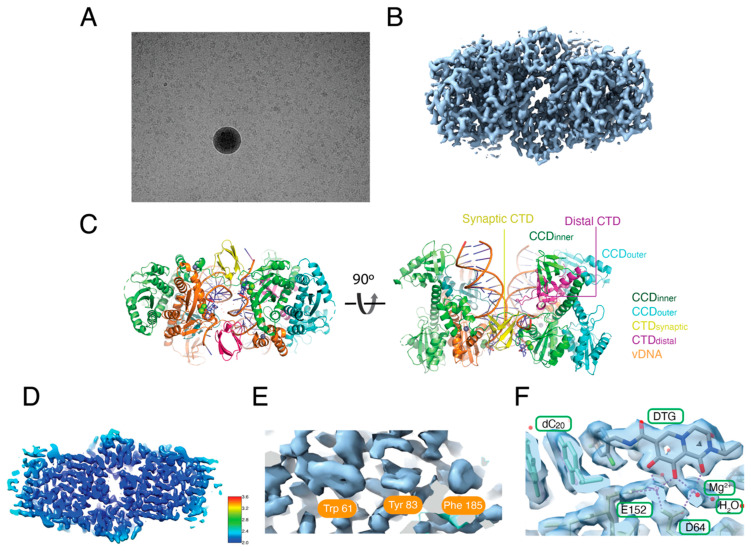
Structure determination of hetero-intasomes. (**A**) Representative micrograph of hetero-intasomes showing mono-dispersed particles. (**B**) Final reconstructed cryo-EM map of the CIC. (**C**) Atomic model of the HIV-1 CIC. The chains are color-coded according to the labels on the right, with synaptic and distal CTDs indicated. (**D**) Local resolution estimation of cryoEM map by ResMap. The sliced map is colored according to the local resolution of the masked intasome core map. (**E**) Representative density showing tryptophan 61, tyrosine 83, and phenylalanine 185 residues in the CCD. It is noted that the features of aromatic ring holes are expected for a 2 Å cryo-EM map. (**F**) Density map around the IN active site and inhibitor binding coordinates, including catalytic residues, D64, E152, bound DTG, Mg^2+^, and H_2_O molecules.

**Figure 5 viruses-16-01166-f005:**
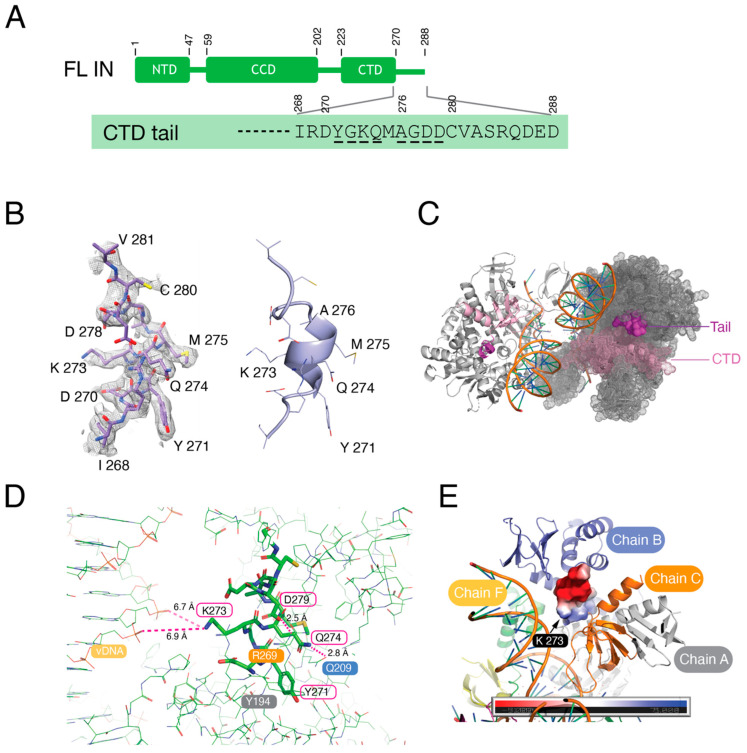
The C-terminal tail stabilizes the intasome core structure. (**A**) Schematic of full-length (FL) HIV-1 IN highlighting C-terminal tail residues 268–288. The residues altered by site-directed mutagenesis are underlined. (**B**) Cryo-EM reconstruction density map (left) and atomic model (right) of the C-terminal tail (residues 268–281). The map is shown with the final structural model superimposed. Residues 271 to 276 form a small alpha helix. (**C**) Distal CTDs (pink) with resolved C-terminal tails 268–281 (purple) are highlighted. The left and right halves of intasome IN protomers are shown as cartoons and dots, respectively. (**D**) Interactions between C-terminal tail residues and vDNA and surrounding IN protomer residues. The C-terminal tail is shown as sticks. The interacting residues are indicated. (**E**) The C-terminal tail (surface colored with electrostatic potential) bridges the interactions of IN protomers. Intasome chains A, B, C, and F are shown as cartoons and are color-coded.

**Figure 6 viruses-16-01166-f006:**
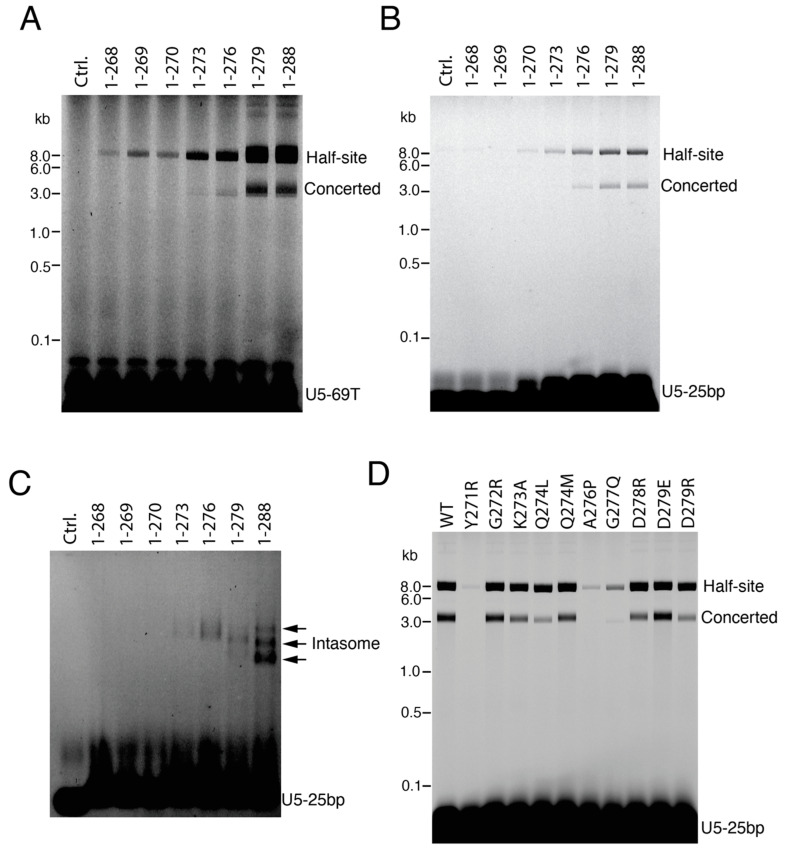
Functional analyses of the IN C-terminal tail. (**A**) WT-IN and C-terminal tail mutant strand transfer activities. The migration positions of half-site and concerted integration reaction products are indicated to the right of the gel image, while the positions of mass standards in kb are shown to the left. (**B**) Strand transfer activities of Sso7d-IN WT and deletion derivatives. (**C**) EMSA detection of intasome assembly. The reaction condition is the same as in panel (**B**), except that 50 μM DTG was added and plasmid target DNA was omitted. Intasomes were analyzed by native 3% agarose gel electrophoresis and detected by a fluorescence scanner. (**D**) Integration assay with Sso7d-IN C-terminal tail missense mutants. The reaction conditions were as in panel B. The results are representative of three independent experiments.

**Figure 7 viruses-16-01166-f007:**
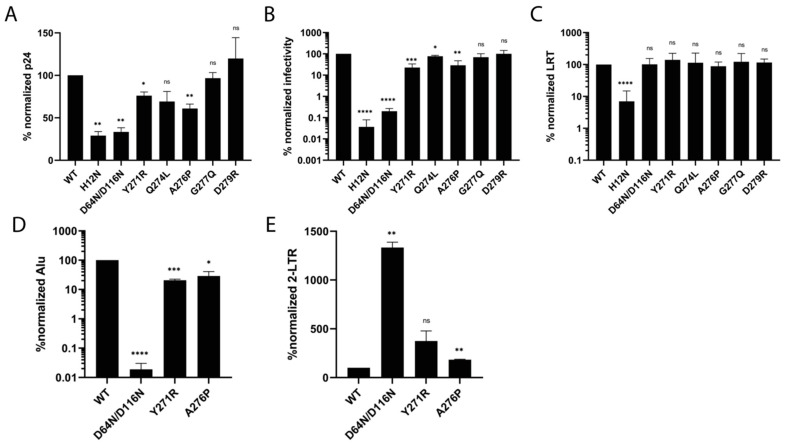
Virus-based experiments. (**A**) Levels of virus release from transfected cells (p24 values; n = 2 independent experiments) were normalized to WT. (**B**) Percent-normalized IN mutant viral infectivities (WT HIV-Luc set to 100%) for n = 2 to 3 independent experiments. (**C**) LRT products were percent-normalized to WT (n = between 3 and 8 independent experiments). (**D**) Alu integration of indicated IN mutant viruses was percent-normalized to WT (n = 2 independent experiments). (**E**) Percent-normalized values of 2-LTR circle formation (n = 2 independent experiments). Results are averages +/− SEM. Statistically significant differences as assessed by two-tailed Student’s *t* test in comparison to WT virus are represented with asterisks (* *p* < 0.05, ** *p* < 0.01, *** *p* < 0.001, **** *p* < 0.0001). ns, not significant.

## Data Availability

The cryo-EM map of the conserved HIV-1 intasome core with DTG was deposited in the Electron Microscopy Data Bank under accession code EMD-45364 while the associated atomic model was deposited in the Protein Data Bank (https://www.wwpdb.org) using PDB code 9C9M.

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
