# Peer review of "HIV-1 Intasomes Assembled with Excess Integrase C-Terminal Domain Protein Facilitate Structural Studies by Cryo-EM and Reveal the Role of the Integrase C-Terminal Tail in HIV-1 Integration"

_viruses, 2024, doi:10.3390/v16071166_

Round 1

Reviewer 1 Report

Comments and Suggestions for Authors

Structural studies about HIV-1 intasomes are limited due to multiple oligomeric nucleocomplexes formed by these retroviral species. This is mostly due to the formation of proto-intasome stacks of octameric oligomers that make difficult to get isolated intasomes for structure studies. Here the authors investigated the hypothesis that addition of integrase CTD during intasome assembly may break the stacks. Data reported in this manuscript indicate clearly that addition of CTD leads to an improved homogeneity of the intasome preparation and increases the resolution of the structures obtained with these preparations. Importantly, this allowed the authors to get structural data for the intasome CTD that were missing in the previous works.

Since these observations will help in the future structural studies and will help in understanding better the role of the integrase CTD in the integration process I would advise to accept the publication of the work. The work presented here appears well performed and I would have only modest points to clarify or discussed before publication.

1-      The stimulation observed in concerted integration assays performed in the presence of added CTD may be also reminiscent of previous stimulations observed with other peptides as LEDGF/p75 (mentioned by the authors) and histone 4 tail peptides. In this cases an increased solubility of the integration complex has been also observed in vitro. Would this stimulation be also due to similar mechanisms? Could the different molecular processes of stimulation by the peptides be additionally discussed?

2-      In the integration assays IN-sso7d shows a higher stimulation of the FSI by the CTD peptide in comparison with HSI, while wild type integrase HSI appears also largely stimulated. In contrast, self integration appears not affected (at the bottom of the gel). How do the authors explain these differences?

3-      Could the authors indicate (or remind) clearly in the MS how the CTD was purified? In which buffer is the CTD prepared? These informations may help in appreciating more precisely the effect of adding CTD in integration assays.

4-      In the EMSA presented in Figure 6, to which complexes correspond the shifted bands migrating at the bottom of the gel? How was determined the migration position of the intasomes?

5-      One major remaining challenge in the field remains to get structural data about the interaction between the intasome and its nucleosomal substrate. Since the CTD could be a partner of the nucleosomal insertion site, how do the authors think this strategy to add CTD may help in these aims?

6-      Integration sites determination may nicely complement the cellular analyzes, do the authors have any clues about this question?

Reviewer 2 Report

Comments and Suggestions for Authors

Li et al report on their discovery of how the addition of excess purified C-terminal domain (CTD) of HIV-1 integrase impacts and improves the assembly of HIV-1 intasomes in vitro, mitigating confounding heterogeneity that occurs during cryo-EM analysis, and subsequently improving the achievable resolution. They observe a small alpha helix not previously observed in residues 271-276 and go on to demonstrate that mutations in this region impact IN activity in vitro and HIV-1 infection in virus.

Overall, this report is very well written, lucid, and thorough – the findings will be an important contribution to the HIV Integrase community collective efforts in this space.

This reviewer’s comments are only minor:

1.      Can the authors comment on their selection of amino acids for the IN(CTD) construct used? Were others tested? For decades, this literature had defined the CTD more narrowly (~220-280), so this is an interesting departure. It would be interesting to know if the additional alpha helical region between 185-220 contributes as well to the observed effect.

2.      Figure 3D&E – please add more annotations to these two gels to help the reader more immediately understand the identity of each lane in the respective experiments?
